# Position Specific Running Performances in Professional Football (Soccer): Influence of Different Tactical Formations

**DOI:** 10.3390/sports8120161

**Published:** 2020-12-10

**Authors:** Toni Modric, Sime Versic, Damir Sekulic

**Affiliations:** 1Faculty of Kinesiology, University of Split, 21000 Split, Croatia; toni.modric@yahoo.com (T.M.); sime.versic@gmail.com (S.V.); 2HNK Hajduk, 21000 Split, Croatia

**Keywords:** team sports, match, tactics, physical capacities, strength and conditioning

## Abstract

Running performances (RPs) are known to be important parameters of success in football (soccer), but there is a lack of studies where RPs are contextualized regarding applied tactical solutions. This study aims to quantify and analyze the differences in position-specific RPs in professional football, when games are played with three defensive players (3DP) and four defensive players (4DP). The participants here include professional football players (M ± SD, age 23.57 ± 2.84 years, body height 181.9 ± 5.17 cm, body mass 78.36 ± 4.18 kg) playing at the highest competitive level in Croatia. RPs were measured by global positioning system and classified into four groups based on playing positions: central defenders (CD; n = 47), wide defenders (WD; n = 24), midfielders (MF; n = 48), or forwards (FW; n = 19). Analysis of variance and discriminant canonical analysis are used to identify differences between 3DP and 4DP tactical solutions in terms of the RPs for each playing position. The number of accelerations and decelerations most significantly contributed to the differentiation of 3DP and 4DP among MFs (Wilks λ = 0.31, *p* < 0.001), with higher occurrences with 3DP. For CDs, total distance, and high-intensity running were higher in 3DP (Wilks λ = 0.66, *p* < 0.001). No multivariate differences were found for FW and WD players in terms of the RPs between 3DP and 4DP tactical formations. The characteristics and differences shown in this study may provide useful information for coaching staff regarding changing in-season tactical formations. Additionally, the results are useful for optimizing training programs for football players with different playing positions. When changing from 4DP to 3DP tactical formations, WDs training programs should include more of high-intensity running, while MFs training programs should be more based on short intensity activities (accelerations and decelerations).

## 1. Introduction

The technical and tactical nature of football (soccer) has resulted in the nature of multifactorial physical characteristics for players [1]. As the nature of the game has evolved over time, the physiological demands have also changed [2,3]. At present, elite football players travel 9 to 14 km in total during a game with high-intensity running accounting for 5–15% of this distance [4,5,6]. These performances vary according to the different playing positions of the players in the game [7,8,9]. Specifically, central midfielders cover the highest overall distance in official football games, while external players (i.e., wingers and fullbacks) cover the greatest distance in terms of high-intensity running [10,11]. It is generally accepted that these differences in running performances (RPs) between playing positions are influenced by the different tactical roles of players during the game [7,12,13].

The team tactical system and individual tactical roles of players (i.e., the positioning and distribution of the players on the pitch) are considered to be among the most important strategical (tactical) decisions in football [14,15,16]. Since their tactical roles are different in different tactical formations [17,18], players consequently experience different physical demands in different tactical formations. Respecting this, training programs should be adapted accordingly. Thus, better understanding of position-specific demand could provide a useful insight to optimize training programs. Therefore, it is important to understand how RPs may be affected by different playing positions in various tactical systems [19]; however, the role of the tactical system regarding the player RP has not been fully described yet and there is a clear lack of information regarding the influence of specific tactical formations on RPs among elite football players [20,21]. To the best of our knowledge, only a few studies have investigated this issue. Bradley et al. investigated team formations in the English Premier League and demonstrated that defenders playing in a 4-4-2 formation covered greater distances compared to those playing in a 4-3-3 or 4-5-1 formation [22]. In a more recent study, Aquino et al. compared the differences between 4-3-3 and 4-4-2 formations and concluded that the RPs were higher for all variables in the 4-3-3 formation when compared to the 4-4-2 formation [23].

Although providing important information about possible associations between tactical solutions and RPs in football, the previously cited studies have investigated team tactical formations with four defensive players (4DP). The 4DP tactical formations are undoubtedly the most common tactical formations in top-level football. However, we are witnessing the growing trend of using tactical formations with three defensive players (3DP) [21]. For instance, in the semi-final of the Champions League of season 2019/2020, two teams played with 3DP (Lyon and RB Leipzig) and two teams with 4DP (Paris Saint Germain and Bayern Munich). In addition, as football practitioners that are deeply involved in elite-level football practice, the authors of the study may say that many football teams nowadays are changing their tactical formations from 4DP to 3DP, and we anticipate that this trend will continue. Therefore, it is important to gain insights into RPs, specifically in terms of the differences between the 4DP and 3DP tactical formations.

To the best of our knowledge, only one study has analyzed RP differences between the 4DP and 3DP tactical formations. Specifically, Baptista et al. recently compared RPs in the 4-5-1 and 3-5-2 tactical systems, highlighting that the general RPs do not differ considerably between these two tactical formations when compared by playing position [21]. In detail, the most relevant exceptions were the higher number of high-intensity accelerations for central defenders in the 4-5-1 tactical formation and the greater high-intensity distances covered by wide defenders in the 3-5-2 tactical formation; however, the cited study only analyzed home games, which limits the generalizability of the presented results to some extent. In general, the home advantage in team sports has an important role in determining team performance [24], and the results from the mentioned study could be influenced by this issue. As a result, it would be important to conduct analysis of RPs for different tactical systems with both home matches and away matches. In addition, the cited study observed the Norwegian league and the matches were played on artificial turf/grass, and therefore, further analyses in other competitions are needed [21]. Therefore, the main objective of this study is to quantify the position-specific RPs of football players in the 3DP and 4DP tactical formations in elite-level football games. The authors believe that the in-depth analysis of RPs across playing positions in different tactical formations could provide (i) valuable information for coaching staff when changing in-season team tactical formations and (ii) useful insights to optimize training programs for football players in different playing positions.

## 2. Materials and Methods

### 2.1. Participants and Design

The participants of this study (n = 20) were professional football players (M ± SD, age 23.57 ± 2.84 years, body height 181.9 ± 5.17 cm, body mass 78.36 ± 4.18 kg) and all of them signed an informed consent to participate in the study. Player RPs were analyzed in 17 games in the Croatian first division during 2018/2019, resulting in the retrieval of 138 RPs which were used as the cases for this study. RPs were observed according to the tactical formations. The 3DP tactical formations were used in 10 games and included the 3-5-2 and 3-4-1-2 formations (Figure 1).

The 4DP tactical formations were used in seven games and included the 4-4-2 (diamond) and 4-1-3-2 formations (Figure 2).

In addition, RPs were classified into four groups based on playing positions: central defenders (CD; n = 47), wide defenders (WD; n = 24), midfielders (MF; n = 48), or forwards (FW; n = 19). All data were obtained with a global positioning system technique (see later for details), and, for the purpose of this study, only the results of players who participated in the whole game were analyzed. In all played games, the team studied here was managed by the same coaching staff. The same training approaches were applied, regardless of playing in different tactical formations. At the end of the observed period, the team was ranked 5th out of 10 teams. The investigation was approved by Ethical Board of the University of Split, Faculty of Kinesiology, Split, Croatia (approval number: 2181-205-02-05-19-0020).

### 2.2. Procedures

The variables in this study included the RPs, player age, body height, and mass (measured by standardized techniques at the beginning of the season). RP was observed according to the different tactical formations and specific playing positions. The 3DP tactical formations consisted of 3 CD, 2 WD, 3 MF, and 2 FW. The 4DP tactical formations consisted of 2 CD, 2 WD, 4 MF, and 2 FW. RP included the total distance covered during the game (m), distances covered in different speed categories (m, i.e., high-speed running (19.8–25.1 km/h), sprinting (≥25.2 km/h), and high-intensity running (>19.8 km/h)), total accelerations (>0.5 m/s^2^), high-intensity accelerations (>3 m/s^2^), total decelerations (≤0.5 m/s^2^), and high-intensity decelerations (≤3 m/s2). RP were measured by Global Positioning System GPS technology (S7 Vector, Catapult, Melbourne, Australia) with a sampling frequency of 10 Hz. During the matches, players wore GPS vests, in which was placed GPS unit that was turned on 15 min before the start of the match. To limit inter-unit error, each player wore the same unit in every match. The reliability and validity of the such equipment has previously been presented in detail [25,26].

### 2.3. Statistics

The normality of the distributions was confirmed by the Kolmogorov–Smirnov test, and the data are presented as the means ± standard deviations. Homogeneity was checked by Levene’s test. Univariate differences in RPs between the 3DP and 4DP tactical formations were analyzed by one-way analysis of variance. To evaluate the effect sizes (ES), partial eta-squared values were found (>0.02 is small; >0.13 is medium; >0.26 is large) [27]. Multivariate differences in RPs between 3DP and 4DP tactical formations were analyzed by canonical discriminant analysis. For all analyses, Statistica 13.0 (TIBCO Software Inc., Greenwood Village, CO, USA) was used, and a *p* < 0.05 was applied.

## 3. Results

The descriptive values and differences within playing positions between the 3DP and 4DP formations for RPs are presented in Table 1 and Table 2. For CDs, significant differences were found for the high-intensity distance covered (529 m with 3DP and 404 m with 4DP; small effect size) and high-intensity decelerations (35 with 3DP and 27 with 4DP, small effect size). For WDs, significant differences were found for the total distance covered (11,021 m with 3DP and 10,143 m with 4DP; large effect size), high-speed distance covered (729 m with 3DP and 505 m with 4DP, large effect size), high-intensity distance covered (955 m with 3DP and 708 m with 4DP, large effect size), and total number of accelerations (485 with 3DP and 451 with 4DP, medium effect size). For MFs, significant differences were found for the total number of accelerations (520 with 3DP and 423 with 4DP; large effect size), total number of decelerations (514 with 3DP and 470 with 4DP; large effect size), high-speed distance covered (632 m with 3DP and 525 m in 4DP, small effect size), and number of high-intensity accelerations (36 with 3DP and 30 with 4DP, small effect size). No differences were found for all FW players in terms of the RPs between 3DP and 4DP tactical formations.

The multivariate differences between the 3DP and 4DP formations that were analyzed by the discriminant analysis showed significant differentiation between the observed tactical solutions for CD (Can R = 0.82, Wilks’ λ = 0.31, *p* < 0.001) and MF (Can R = 0.82, Wilks’ λ = 0.31, *p* < 0.01). The accelerations (correlation with discriminant function (r) = 0.87) and decelerations (r = 0.84) numbers most significantly contributed to the differentiation of MF, with higher occurrences for both RPs with 3DP. A discriminant function correctly classified 83% of cases (86% and 79% for 3DP and 4DP, respectively). For CD, total distance (r = 0.78) and high-intensity running (r = 0.55) were highly correlated with discriminant function, indicating importance of these variables in discriminating 3DP and 4DP with regard to RPs. Supportively to previously presented ANOVA results, total distance and high-intensity running were higher with 3DP, while the discriminant root correctly classified 96% of observed cases (93% and 100% for 3DP and 4DP, respectively) (Table 3).

## 4. Discussion

The main objective of this study was to determine the differences in RPs between the 3DP and 4DP tactical formations within playing positions in official elite-level football games. In general, the results indicate that the values for almost all of the RP metrics are greater with 3DP than 4DP. Specifically, in 3DP tactical formations, CDs featured greater high-intensity running than in 4DP tactical formations. Second, WDs had a greater total distance covered in 3DP tactical formations and played with more high-intensity running when compared to 4DP tactical formations. Finally, MFs had greater numbers of total accelerations and decelerations, as well as greater high-speed running in 3DP than in 4DP tactical formations, but the RPs did not differ significantly between 3DP and 4DP for FWs.

### 4.1. Central Defenders

CDs featured more high-intensity running in 3DP tactical formations than in 4DP tactical formations (529 ± 180 and 404 ± 138 m, respectively). In addition, all RP indicators were numerically higher in 3DP formations. Considering the different roles of CDs in 3DP and 4DP tactical formations [17,18], these results are not surprising. In particular, when the build-up of the attack starts from the goalkeeper in 3DP formations (i.e., 3-5-2, 3-4-1-2, or 3-4-3), CDs are more involved in attacking actions than in 4DP tactical formations. In such cases, the CDs are positioned wider and deeper in the opponent’s half of the pitch. Such positioning allows them more potential options for key passes directed toward FW, WD, and MF players; however, because of such positioning (wider and deeper in the opponent’s half), CDs leave greater space behind their back in 3DP formations than in 4DP formations.

Meanwhile, defending the space left behind is one of the most important defensive duties of defensive players (e.g., CD and WD players). In particular, lost ball possession is regularly followed by a rapid offensive transition by the opposing team (i.e., a counterattack). To defend successfully, CDs must return to their starting position in defense. While running back, maximal and submaximal intensities are typically reached. This includes high-intensity running (i.e., high-speed running and sprinting). Therefore, during this phase of the game, CDs achieve the majority of their total amount of high-intensity running. Considering the above, it is logical that CDs feature greater high-intensity running in 3DP tactical formations than in 4DP tactical formations.

### 4.2. Wide Defenders

WDs experienced a greater total distance covered and 30% more of high-intensity running in 3DP when compared to 4DP tactical formations. Such results can be explained by the different roles of WDs in 3DP and 4DP tactical formations [17,18].

First, in 3DP tactical formations, WDs are positioned much deeper in the opponent’s half than in 4DP tactical formations. In addition, 3DP tactical formations use three CDs, while 4DP tactical formations use two CDs. This actually means that 3DP tactical formations include additional players in defense when compared to 4DP formations. Consequently, WDs in 3DP formations are more easily able to contribute to attacks and participate in offensive actions. In accordance with this, Riboli et al. recently presented the results of a study where they investigated the effects of the playing formation on physical demands in elite-level football, concluding that wide defenders showed greater physical demands in the 3-5-2 tactical formation than in other observed tactical formations (e.g., 4-4-2, 4-3-3, 3-4-1-2, or 3-4-2-1) [28].

On the other hand, in defensive actions, WDs travel deeply into their own half of the pitch, and, together with the three CDs, form a line of five defensive players. Basically, WDs in 3DP tactical formations must participate equally in both defensive and offensive actions. This is probably one of the most important reasons for the large difference in total distance and high-intensity running (high-speed and sprint running) between the 3DP and 4DP tactical formations for WDs. A similar trend was highlighted in a recent study where Baptista et al. compared physical demands in matches between 4-5-1 and 3-5-2 tactical formations with professional football players from an elite-level Norwegian division [21]. Although no statistically significant differences were found, the descriptive parameters indicated slightly higher values for the total distance covered and high-intensity running for WDs in a 3-5-2 tactical formation when compared to a 4-5-1 tactical formation. Despite some differences (the Norwegian study exclusively analyzed home games played on artificial turf/grass, while herein we observed both home and away matches played on grass), the results are supportive to the conclusion of higher physical demands for WDs in 3DP tactical formations.

### 4.3. Midfielders

Present study included the 3-5-2 and 3-4-1-2 tactical formations. These tactical formations consist of 3 CD, 2 WD, 3 MF, and 2 FW. On the other hand, the 4-4-2 (diamond) and 4-1-3-2 tactical formations (4DP formations included here) consist of 2 CD, 2 WD, 4 MF, and 2 FW. Thus, 3DP tactical formations generally have minimally one player less in the midfield than 4DP tactical formations. Almost certainly, the reduced number of players in the midfield was compensated by the greater activity of MF within the 3DP system, resulting in the higher number of accelerations and decelerations with 3DP. Additionally, our results indicate that MFs in 3DP tactical formations feature a slightly increased total distance covered and 15–20% higher values for high-intensity running and high-speed running. It seems that due the reduced number of midfield players in 3DP tactical formations, MFs cover greater space in 3DP formations and play at a higher game pace than in 4DP formations.

It is well known that MFs are key players for organizing offensive actions [29]. Therefore, we believe that our results could improve their training process and consequently affect the quality of the game organization within the whole team. In particular, training drills for MFs which play in 3DP tactical formations should primarily consist a large number of changes of direction and repeated accelerations and decelerations. As a result, MF players included in 3DP tactical formations should have a better overall conditioning status then MFs which play in 4DP tactical formations.

### 4.4. Forwards

No significant differences were found for players that played at the FW position. This may seem surprising, but this finding is understandable. In brief, irrespective of the differences between the 3DP and 4DP formations in terms of the number of defensive of players (see previous text for details), the tactical formations observed in this study (3-5-2, 3-4-1-2, and 4-4-2 diamond, 4-1-3-2) always included two FW. Therefore, the roles of the FW players were very similar, irrespective of the tactical formation. This probably explains the similar values in terms of the RPs between different tactical formations. Thus, although RPs during football matches vary due to different tactical roles [11], this was not the case for the FW players that were analyzed in this study.

### 4.5. Limitations and Strengths

The main limitation of the study comes from the fact that players who competed in the Croatian national league were observed, and therefore the results are only generalizable to samples who compete at a similar level. In addition, the playing positions were unequal in terms of their sample sizes, which almost certainly influenced the possibility of finding statistically significant differences within certain playing positions. Finally, the influence of opponent quality on RP must not be underestimated, and, therefore, it may have influenced the presented results to some extent.

This is one of the first studies where professional players were observed, and where 3DP and 4DP tactical solutions were compared with regard to RP, which is an important strength of this investigation. In addition, to the best of our knowledge, this is the first study where both home and away performances in professional football were included in the analyses. Therefore, although not being the final word on a problem, we hope that our results will improve the knowledge and initiate further research.

### 4.6. Practical Applications

Considering that high intensity distance covered in matches is closely related to training status [30], the findings of this study may help coaches to identify WD players who will be able to respond to the physical demands during matches played with 3DP tactical formations. In addition, exposing players to large and rapid increases of high intensity distances (i.e., when changing from 3DP to 4DP) increase the rates of injuries. Therefore, specific injury prevention programs are highly recommended to be applied in training programs of WDs.

Due to the different organizations of players at 3DP and 4DP, MF players should be far more active in 3DP tactical formations than in 4DP tactical formations. For this particular playing position, the differences are mostly evident in the numbers of accelerations and decelerations, with higher demands in 3DP. From the perspective of football-specific conditioning, it is important to use these findings when creating position-specific training drills for MF players.

In the context of the possible change of tactical formations from 4DP to 3DP, it should not be difficult for CDs to adapt in terms of the RP changes. In addition, the findings of this study can facilitate decisions for football coaches when selecting FW players. In particular, if a change of tactical formations does not include a change in the number of players in attack, FW players will probably be able to respond to the physical demands of their position, regardless of the number of players in defense or midfield. On the other hand, it is questionable whether FW players will be able to respond to the changing physical demands when changing the tactical formations leads to a change in the number of players in the FW position. Therefore, future research should investigate whether RPs differ in tactical formations when teams play with one and two FW players (e.g., 4-2-3-1 vs 3-5-2).

## 5. Conclusions

This study demonstrated that a possible change in tactical formation from 4-4-2 (diamond) or 4-1-3-2 to 3-5-2 or 3-4-1-2 would differently affect the RPs of players at different playing positions. In particular, the RPs of WD players differ the most, indicating greater total distance covered and high intensity distance covered in 3DP formation. The RPs of MF players are also substantially different between the 3DP and 4DP formations. Specifically, the differences are mostly evident in the numbers of accelerations and decelerations, with higher demands in the 3DP formation. Slightly increased values for all RPs, with emphasis on high-intensity running, were observed for CD players in 3DP tactical formations; however, the differences between the 3DP and 4DP formations for CD are much lower than for WD and MF players.

These findings may help coaches to identify which players would be able to respond to the physical demands during matches played with 3DP tactical formations. In addition, from the perspective of football-specific conditioning, it is important to use these findings when creating position-specific training drills for players on specific playing positions.

## Figures and Tables

**Figure 1 sports-08-00161-f001:**
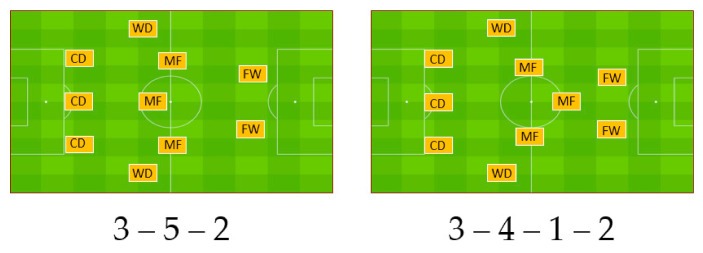
Tactical systems used in formations with three defensive players (CD—central defenders, WD—wide defenders, MF—midfielders, FW—forwards).

**Figure 2 sports-08-00161-f002:**
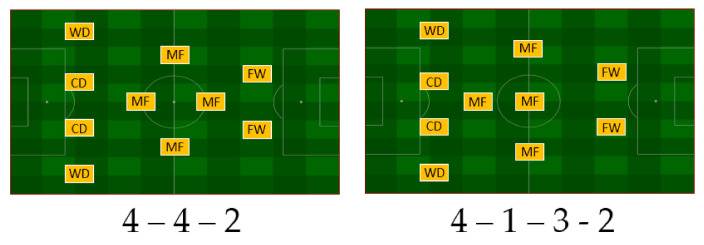
Tactical systems used in formations with four defensive players (CD—central defenders, WD—wide defenders, MF—midfielders, FW—forwards).

**Table 1 sports-08-00161-t001:** Descriptive statistics for tactical formations involving three- (3DP) and four-defensive players (4DP).

Variables	Tactical Formation	CD	WD	MF	FW
Total distance covered	3DP	10,327.4 ± 536.6	11,021.6 ± 361.8	11,731.0 ± 519.7	10,501.2 ± 647.0
	4DP	10,106.9 ± 472.0	10,143.7 ± 434.7	11,418.6 ± 549.6	9827.6 ± 757.7
High-intensity distance	3DP	529.0 ± 180.4	955.1 ± 160.9	725.9 ± 205.5	915.8 ± 185.8
	4DP	404.3 ± 139.1	708.2 ± 151.8	628.2 ± 138.6	795.6 ± 213.6
High speed distance	3DP	428.7 ± 130.9	729.3 ± 89.8	632.5 ± 164.6	696.1 ± 149.8
	4DP	332.9 ± 103.4	505.2 ± 86.7	525.2 ± 105.3	605.3 ± 145.0
Sprint distance	3DP	100.2 ± 57.7	226.0 ± 101.8	93.5 ± 53.8	219.9 ± 71.5
	4DP	71.3 ± 48.2	203.0 ± 92.7	103.1 ± 45.9	190.8 ± 82.9
Total accelerations	3DP	481.7 ± 51.7	485.9 ± 34.1	520.5 ± 28.6	414.7 ± 110.9
	4DP	467.1 ± 42.6	451.2 ± 42.6	473.4 ± 37.1	415.8 ± 38.0
Total decelerations	3DP	479.2 ± 51.7	476.3 ± 33.7	514.9 ± 24.4	440.3 ± 42.2
	4DP	465.7 ± 39.0	448.1 ± 44.4	470.1 ± 40.2	415.8 ± 39.8
High intensity accelerations	3DP	22.3 ± 10.4	24.8 ± 9.6	19.5 ± 9.0	37.9 ± 16.0
	4DP	19.1 ± 4.8	28.0 ± 5.2	21.1 ± 6.5	40.9 ± 8.2
High intensity decelerations	3DP	35.6 ± 10.7	43.1 ± 11.1	36.5 ± 10.4	52.7 ± 11.6
	4DP	27.8 ± 8.7	37.7 ± 6.5	30.7 ± 5.1	48.9 ± 10.1

CD = central defenders, WD = wide defenders, MF = midfielders, FW = forwards; 3DP = 3 defensive players tactical formations, 4DP = 4 defensive players tactical formations.

**Table 2 sports-08-00161-t002:** Differences within playing positions between tactical formations involving three and four defensive players (analysis of variance—ANOVA).

Variables	ANOVA	CD	WD	MF	FW
Total distance covered	F-test	1.94	29.08	3.90	4.29
	*p*	0.17	0.01	0.05	0.05
	η^2^	0.04	0.57	0.07	0.20
High-intensity distance	F-test	5.87	14.39	3.27	1.69
	*p*	0.02	0.01	0.08	0.21
	η^2^	0.11	0.39	0.07	0.09
High speed distance	F-test	6.50	37.37	6.30	1.80
	*p*	0.01	0.01	0.02	0.20
	η^2^	0.12	0.63	0.12	0.09
Sprint distance	F-test	2.97	0.32	0.40	0.66
	*p*	0.09	0.58	0.53	0.43
	η^2^	0.06	0.01	0.00	0.04
Total accelerations	F-test	0.95	4.90	24.08	0.00
	*p*	0.33	0.04	0.01	0.98
	η^2^	0.02	0.18	0.35	0.00
Total decelerations	F-test	0.84	3.14	22.71	1.70
	*p*	0.36	0.09	0.01	0.21
	η^2^	0.01	0.12	0.33	0.09
High intensity accelerations	F-test	1.35	0.91	0.43	0.28
	*p*	0.25	0.35	0.51	0.61
	η^2^	0.02	0.03	0.00	0.01
High intensity decelerations	F-test	6.26	1.87	5.05	0.57
	*p*	0.02	0.19	0.03	0.46
	η^2^	0.12	0.07	0.10	0.03

CD = central defenders, WD = wide defenders, MF = midfielders, FW = forwards.

**Table 3 sports-08-00161-t003:** Multivariate differences between tactical formations involving three- (3DP) and four-defensive players (4DP) in running performances defined by discriminant canonical analysis *.

Position	CD	WD	MF	FW
Variables	Root	Root	Root	Root
Total distance covered	0.78	0.43	−0.35	−0.57
High-intensity distance	0.55	0.75	−0.32	−0.36
Total accelerations	0.32	0.30	−0.87	0.01
Total decelerations	0.26	0.29	−0.84	−0.36
High intensity accelerations	−0.14	0.36	0.12	0.14
High intensity decelerations	0.20	0.78	−0.40	−0.21
Can R	0.82	0.42	0.64	0.66
Wilks λ	0.31	0.81	0.58	0.56
*p*	0.00	0.19	0.00	0.23
Centroid: 3DP	1.19	0.32	−0.67	−0.88
Centroid: 4DP	−1.67	−0.65	1.00	0.79

CD = central defenders, WD = wide defenders, MF = midfielders, FW = forwards; Can R—canonical correlation, root—structure of the discriminant function/root. * Discriminant analysis did not include high speed distance and sprint distance covered because these variables summarized equals high-intensity distance which was included in analysis.

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
