# Peer review of "Position Specific Running Performances in Professional Football (Soccer): Influence of Different Tactical Formations"

_sports, 2020, doi:10.3390/sports8120161_

Round 1

Reviewer 1 Report

This study has highlighted some interesting results. It is clear and straightforward to follow. The main criticism I have is that the authors sometimes repeat themselves in the text which resulted in having a long discussion and conclusion. Below are my points in more details:

  • Line 42-46 needs to be explained better. Here the authors should explain in more details of why the carried out this study.
  • The authors argued that formations with 3DP are getting more popular and they basically based the whole study on this hypothesis. Can the authors provide more evidence for this? 
  • Line 108-109 the authors mentioned that "only the results of players who participated in the whole game were analysed". Is there any explanation of why you did not count the substitute players in your study? Also, how did that influenced your sample size?
  • With 3DP formation being more physiologically demanding for players, was there any correlation when 3DP or 4DP was selected and the rate of injury in the players during the league? If this has not been done, can the authors discuss and cite some other studies which have done so.
  • I found the results section to be very brief and the discussion way too long. I suggest moving the first paragraphs of 4.1- 4.4 to the results section and modify it accordingly.
  • The conclusion is too wordy and I suggest to summarize only the highlights of the study in this section.

Author Response

Reviewer 1

Comments and Suggestions for Authors

This study has highlighted some interesting results. It is clear and straightforward to follow. The main criticism I have is that the authors sometimes repeat themselves in the text which resulted in having a long discussion and conclusion. Below are my points in more details:

Thank you for recognizing potential of our manuscript. We will try to follow all your comments and amended manuscript accordingly.

  • Line 42-46 needs to be explained better. Here the authors should explain in more details of why the carried out this study.

Thank you for this suggestion. We amended manuscript accordingly, and now that part reads:

The team tactical system and individual tactical roles of players (i.e., the positioning and distribution of the players on the pitch) are considered to be among the most important strategical (tactical) decisions in football [14-16]. Since tactical roles are different in different tactical formations [17, 18], it is evident that players experience different physical demands in different tactical formations. Respecting this, training programs of players should be adapted accordingly. Thus, better understanding of these position-specific demands could provide a useful insight to optimize training programs. Therefore, it is important to understand how running performances may be affected by different playing positions in various tactical systems [19]; however, the role of the tactical system regarding the player running performance has not been fully described yet and there is a clear lack of information regarding the influence of specific tactical formations on running performances among elite football players [20, 21]. To the best of our knowledge, only a few studies have investigated this issue. Bradley et al. investigated team formations in the English Premier League and demonstrated that defenders playing in a 4-4-2 formation covered greater distances compared to those playing in a 4-3-3 or 4-5-1 formation [22]. In a more recent study, Aquino et al. compared the differences between 4-3-3 and 4-4-2 formations and concluded that the running performances were higher for all variables in the 4–3–3 formation when compared to the 4–4–2 formation [23].”

  • The authors argued that formations with 3DP are getting more popular and they basically based the whole study on this hypothesis. Can the authors provide more evidence for this? 

Thank you for this suggestion. We provided evidence for this issue, text now reads: “Although providing important information about possible associations between tactical solutions and running performances in football, the previously cited studies have investigated team tactical formations with four defensive players (4DP), which is undoubtedly the most common tactical formation in modern football; however, we are witnessing the growing trend of using tactical formations with three defensive players (3DP) [21]. For instance, according to UEFA official reports, in Semi-final of Champions league of season 2019/2020, two teams played with 3DP (Lyon and RB Leipzig) and two teams with 4DP (Paris Saint Germain and Bayern Munich). Also, as football practitioners that are deeply involved in elite-level football practice, the authors of the study may say that many football teams nowadays are changing their tactical formations from 4DP to 3DP, and we anticipate that this trend will continue. Therefore, it is important to gain insights into running performances, specifically in terms of the differences between the 4DP and 3DP tactical formations.”

  • Line 108-109 the authors mentioned that "only the results of players who participated in the whole game were analysed". Is there any explanation of why you did not count the substitute players in your study? Also, how did that influenced your sample size?

We did not count substitute players because it is very hard to compare running performances of players that play 90 minutes with players that played, for example, 20 minutes. Since we analysed absolute values of running performances, basically all running performances of substitute players will be lower than players that played 90 minutes (simply because they spent less of time in the game).

In the end, we have followed methodology of previous works that analysed running performances in official soccer matches. For example:

Baptista, I., et al., A comparison of match-physical demands between different tactical systems: 1-4-5-1 vs 1-3-5-2. PLOS ONE, 2019. 14(4): p. e0214952

Modric, T., et al., Analysis of the Association between Running Performance and Game Performance Indicators in Professional Soccer Players. International journal of environmental research and public health, 2019. 16(20): p. 4032

  • With 3DP formation being more physiologically demanding for players, was there any correlation when 3DP or 4DP was selected and the rate of injury in the players during the league? If this has not been done, can the authors discuss and cite some other studies which have done so.

Unfortunately, we did not establish correlation between tactical formations and the rate of injury in the players during the league.

According to your suggestions, we cited study which demonstrated that exposing players to large and rapid increases of high intensity distances (i.e., when changing from 3DP to 4DP) increase the rates of injuries. This is especially important for WD players which experience 30% greater high intensity distance covered. Thank you for this suggestion, please find text which reads:

“Finally, since exposing players to large and rapid increases of high intensity distances (i.e., when changing from 3DP to 4DP) increase the rates of injuries, injury prevention programs are highly recommended to be applied in training programs of WDs.”

  • I found the results section to be very brief and the discussion way too long. I suggest moving the first paragraphs of 4.1- 4.4 to the results section and modify it accordingly.

We systematically re-wrote Discussion according to your comments. Discussion is now reduced, and text reads:

“4. Discussion

The main objective of this study was to determine the differences in RPs between the 3DP and 4DP tactical formations within playing positions in official elite-level football games. In general, the results indicate that the values for almost all of the RP metrics are greater with 3DP than 4DP. Specifically, in 3DP tactical formations, CDs featured greater high-intensity running than in 4DP tactical formations. Second, WDs had a greater total distance covered in 3DP tactical formations and played with more high-intensity running when compared to 4DP tactical formations. Finally, MFs had greater numbers of total accelerations and decelerations, as well as greater high-speed running in 3DP than in 4DP tactical formations, but the RPs did not differ significantly between 3DP and 4DP for FWs.

4.1 Central defenders

CDs featured more high-intensity running in 3DP tactical formations than in 4DP tactical formations (529 ± 180 and 404 ± 138 m, respectively). In addition, all RP indicators were numerically higher in 3DP formations. Considering the different roles of CDs in 3DP and 4DP tactical formations [17, 18], these results are not surprising. In particular, when the build-up of the attack starts from the goalkeeper in 3DP formations (i.e., 3-5-2, 3-4-1-2, or 3-4-3), CDs are more involved in attacking actions than in 4DP tactical formations. In such cases, the CDs are positioned wider and deeper in the opponent's half of the pitch. Such positioning allows them more potential options for key passes directed toward FW, WD, and MF players; however, because of such positioning (wider and deeper in the opponent’s half), CDs leave greater space behind their back in 3DP formations than in 4DP formations.

Meanwhile, defending the space left behind is one of the most important defensive duties of defensive players (e.g., CD and WD players). In particular, lost ball possession is regularly followed by a rapid offensive transition by the opposing team (i.e., a counterattack). To defend successfully, CDs must return to their starting position in defence. While running back, maximal and submaximal intensities are typically reached. This includes high-intensity running (i.e., high-speed running and sprinting). Therefore, during this phase of the game, CDs achieve the majority of their total amount of high-intensity running. Considering the above, it is logical that CDs feature greater high-intensity running in 3DP tactical formations than in 4DP tactical formations.

4.2 Wide defenders

WDs experienced a greater total distance covered and 30% more of high-intensity running in 3DP when compared to 4DP tactical formations. Such results can be explained by the different roles of WDs in 3DP and 4DP tactical formations [17, 18].

First, in 3DP tactical formations, WDs are positioned much deeper in the opponent's half than in 4DP tactical formations. Also, 3DP tactical formations use three CDs, while 4DP tactical formations use two CDs. This actually means that 3DP tactical formations include additional players in defence when compared to 4DP formations. Consequently, WDs in 3DP formations are more easily able to contribute to attacks and participate in offensive actions. In accordance with this, Riboli et al. recently presented the results of a study where they investigated the effects of the playing formation on physical demands in elite-level football, concluding that wide defenders showed greater physical demands in the 3-5-2 tactical formation than in other observed tactical formations (e.g., 4-4-2, 4-3-3, 3-4-1-2, or 3-4-2-1) [28].

On the other hand, in defensive actions, WDs travel deeply into their own half of the pitch, and, together with the three CDs, form a line of five defensive players. Basically, WDs in 3DP tactical formations must participate equally in both defensive and offensive actions. This is probably one of the most important reasons for the large difference in total distance and high-intensity running (high-speed and sprint running) between the 3DP and 4DP tactical formations for WDs. A similar trend was highlighted in a recent study where Baptista et al. compared physical demands in matches between 4-5-1 and 3-5-2 tactical formations with professional football players from an elite-level Norwegian division [21]. Although no statistically significant differences were found, the descriptive parameters indicated slightly higher values for the total distance covered and high-intensity running for WDs in a 3-5-2 tactical formation when compared to a 4-5-1 tactical formation. Despite some differences (the Norwegian study exclusively analysed home games played on artificial turf/grass, while herein we observed both home and away matches played on grass), the results are supportive to the conclusion of higher physical demands for WDs in 3DP tactical formations.

4.3 Midfielders

Present study included the 3-5-2 and 3-4-1-2 tactical formations. These tactical formations consist of 3 CD, 2 WD, 3 MF, and 2 FW. On the other hand, the 4-4-2 (diamond) and 4-1-3-2 tactical formations (4DP formations included here) consist of 2 CD, 2 WD, 4 MF, and 2 FW. Thus, 3DP tactical formations generally have minimally one player less in the midfield than 4DP tactical formations. Almost certainly, the reduced number of players in the midfield was compensated by the greater activity of MF within the 3DP system, resulting in the higher number of accelerations and decelerations with 3DP. Additionally, our results indicate that MFs in 3DP tactical formations feature a slightly increased total distance covered and 15-20% higher values for high-intensity running and high-speed running. It seems that due the reduced number of midfield players in 3DP tactical formations, MFs cover greater space in 3DP formations and play at a higher game pace than in 4DP formations.

It is well known that MFs are key players for organizing offensive actions [29]. Therefore, we believe that our results could improve their training process and consequently affect the quality of the game organization within the whole team. In particular, training drills for MFs which play in 3DP tactical formations should primarily consist a large number of changes of direction and repeated accelerations and decelerations. As a result, MF players included in 3DP tactical formations should have a better overall conditioning status then MFs which play in 4DP tactical formations.

4.4 Forwards

No significant differences were found for players that played at the FW position. This may seem surprising, but this finding is understandable. In brief, irrespective of the differences between the 3DP and 4DP formations in terms of the number of defensive of players (see previous text for details), the tactical formations observed in this study (3-5-2, 3-4-1-2, and 4-4-2 diamond, 4-1-3-2) always included two FW. Therefore, the roles of the FW players were very similar, irrespective of the tactical formation. This probably explains the similar values in terms of the RPs between different tactical formations. Thus, although RPs during football matches vary due to different tactical roles [11], this was not the case for the FW players that were analysed in this study.

4.5 Limitations and strengths

The main limitation of the study comes from the fact that players who competed in the Croatian national league were observed, and therefore the results are only generalizable to samples who compete at a similar level. Also, the playing positions were unequal in terms of their sample sizes, which almost certainly influenced the possibility of finding statistically significant differences within certain playing positions. Finally, the influence of opponent quality on running performance must not be underestimated, and, therefore, it may have influenced the presented results to some extent.

This is one of the first studies where professional players were observed, and where 3DP and 4DP tactical solutions were compared with regard to RP, which is an important strength of this investigation. Also, to the best of our knowledge, this is the first study where both home and away performances in professional football were included in the analyses. Therefore, although not being the final word on a problem, we hope that our results will improve the knowledge and initiate further research.”

4.6 Practical applications

Considering that high intensity distance covered in matches is closely related to training status [30], the findings of this study may help coaches to identify WD players who will be able to respond to the physical demands during matches played with 3DP tactical formations. Also, as the exposing players to large and rapid increases of high intensity distances (i.e., when changing from 3DP to 4DP) increase the rates of injuries, injury prevention programs are highly recommended to be applied in training programs of WDs.

Due to the different organizations of players between 3DP and 4DP, MF players should be far more active in 3DP tactical formations than 4DP tactical formations. For this particular playing position, the differences are mostly evident in the numbers of accelerations and decelerations, with higher demands with the 3DP formation. From the perspective of football-specific conditioning, it is important to use these findings when creating position-specific training drills for MF players.

In the context of the possible change of tactical formations from 4DP to 3DP, it should not be difficult for CDs to adapt in terms of the RP changes. Also, the findings of this study can facilitate decisions for football coaches when selecting FW players. In particular, if a change of tactical formations does not include a change in the number of players in attack, FW players will probably be able to respond to the physical demands of their position, regardless of the number of players in defence or midfield. On the other hand, it is questionable whether FW players will be able to respond to the changing physical demands when changing the tactical formations leads to a change in the number of players in the FW position. Therefore, future research should investigate whether RPs differ in tactical formations when teams play with one and two FW players (e.g., 4-2-3-1 vs 3-5-2).

  • The conclusion is too wordy and I suggest to summarize only the highlights of the study in this section.

Conclusion part is amended according to your suggestions. Now it reads:

“This study demonstrated that a possible change in tactical formation from 4-4-2 (diamond) or 4-1-3-2 to 3-5-2 or 3-4-1-2 would differently affect the RPs of players at different playing positions. In particular, the RPs of WD players differ the most, indicating greater total distance covered and high intensity distance covered in 3DP formation. The RPs of MF players are also substantially different between the 3DP and 4DP formations. Specifically, the differences are mostly evident in the numbers of accelerations and decelerations, with higher demands in the 3DP formation. Slightly increased values for all RPs, with emphasis on high-intensity running, were observed for CD players in 3DP tactical formations; however, the differences between the 3DP and 4DP formations for CD are much lower than for WD and MF players. 

These findings may help coaches to identify which players would be able to respond to the physical demands during matches played with 3DP tactical formations. Also, from the perspective of football-specific conditioning, it is important to use these findings when creating position-specific training drills for players on specific playing positions.”

Reviewer 2 Report

The manuscript “Position specific running performances in 3 professional football (soccer): influence of different 4 tactical formations analysed the differences in position-specific RPs in professional football, when games are played with three defensive players (3DP) and four defensive players (4DP). The information about characteristics and differences found may provide useful information for coaching staff regarding changing in-season tactical formations.

This is a novel and well conducted study, with well-designed methodology and with a direct practical application

Introduction

Well described and developed

Methods

Specified number of subject analyzed

Procedure must be described in deep for better comprehension

Specified statistical program used

Results

Correct

Discussion

Include a practical application section,

Conclusion must respond the study aims, reduce it. Most of the information in this point must be in discussion

Author Response

Reviewer 2

Comments and Suggestions for Authors

The manuscript “Position specific running performances in 3 professional football (soccer): influence of different 4 tactical formations analysed the differences in position-specific RPs in professional football, when games are played with three defensive players (3DP) and four defensive players (4DP). The information about characteristics and differences found may provide useful information for coaching staff regarding changing in-season tactical formations.

This is a novel and well conducted study, with well-designed methodology and with a direct practical application.

Thank you very much for finding our manuscript well-written. We will try to follow all your comments and amended manuscript accordingly.

Introduction

Well described and developed

Thank you.

Methods

Specified number of subject analysed

Exact number of players that participated in this project is added (n = 20) in the first paragraph of Methods. Text reads:

“The participants of this study (n = 20) were professional football players (M ± SD, age 23.57 ± 2.84 years, body height 181.9 ± 5.17 cm, body mass 78.36 ± 4.18 kg) and all of them signed an informed consent form agreeing to participate in the study.”

Procedure must be described in deep for better comprehension

Thank you for this suggestion. We added some details in procedure sections. Text now reads:

“The variables in this study included the RP and player age, body height, and mass (measured by standardized techniques at the beginning of the season). RP was observed according to the different tactical formations and specific playing positions. 3DP tactical formations consist of 3 CD, 2 WD, 3 MF, and 2 FW. 4DP tactical formations consist of 2 CD, 2 WD, 4 MF, and 2 FW. RP and included the total distance covered during the game (m), distances covered in different speed categories (m, i.e., high-speed running (19.8–25.1 km/h), sprinting (≥25.2 km/h), and high-intensity running (>19.8 km/h)), total accelerations (>0.5 m/s2), high-intensity accelerations (>3 m/s2), total decelerations (<–0.5 m/s2), and high-intensity decelerations (<–3 m/s2). RP were measured by GPS technology (S7 Vector, Catapult, Melbourne, Australia) with a sampling frequency of 10 Hz. During the matches players wore GPS vest, in which was placed GPS unit that was turned on 15 minutes before the start of the match. To limit inter-unit error, each player wore the same unit in every match. The reliability and validity of the such equipment has previously been presented in detail [25, 26].

Specified statistical program used

Amended accordingly. We added sentence: “For all analyses, Statistica 13.0 (TIBCO Software Inc., Greenwood Village, CO, USA) was used, and a p < 0.05 was applied”

Results

Correct

Thank you.

Discussion

Include a practical application section

Thank you for this suggestion. Practical application section is now added, please find heading 4.6. Text reads:

“4.6 Practical applications

Considering that high intensity distance covered in matches is closely related to training status [30], the findings of this study may help coaches to identify WD players who will be able to respond to the physical demands during matches played with 3DP tactical formations. Also, as the exposing players to large and rapid increases of high intensity distances (i.e., when changing from 3DP to 4DP) increase the rates of injuries, injury prevention programs are highly recommended to be applied in training programs of WDs.

Due to the different organizations of players between 3DP and 4DP, MF players should be far more active in 3DP tactical formations than 4DP tactical formations. For this particular playing position, the differences are mostly evident in the numbers of accelerations and decelerations, with higher demands with the 3DP formation. From the perspective of football-specific conditioning, it is important to use these findings when creating position-specific training drills for MF players.

In the context of the possible change of tactical formations from 4DP to 3DP, it should not be difficult for CDs to adapt in terms of the RP changes. Also, the findings of this study can facilitate decisions for football coaches when selecting FW players. In particular, if a change of tactical formations does not include a change in the number of players in attack, coaches can be sure that FW players will be able to respond to the physical demands of their position, regardless of the number of players in defence or midfield. On the other hand, it is questionable whether FW players will be able to respond to the changing physical demands when changing the tactical formations leads to a change in the number of players in the FW position. Therefore, future research should investigate whether RPs differ in tactical formations when teams play with one and two FW players (e.g., 4-2-3-1 vs 3-5-2).”

Conclusion must respond the study aims, reduce it. Most of the information in this point must be in discussion

Thank you for this suggestion. We reduced Conclusion and most of the information moved to the Discussion and Practical application parts. Text now reads:

“This study demonstrated that a possible change in tactical formation from 4-4-2 (diamond) or 4-1-3-2 to 3-5-2 or 3-4-1-2 would differently affect the RPs of players at different playing positions. In particular, the RPs of WD players differ the most, indicating greater total distance covered and high intensity distance covered in 3DP formation. The RPs of MF players are also substantially different between the 3DP and 4DP formations. Specifically, the differences are mostly evident in the numbers of accelerations and decelerations, with higher demands in the 3DP formation. Slightly increased values for all RPs, with emphasis on high-intensity running, were observed for CD players in 3DP tactical formations; however, the differences between the 3DP and 4DP formations for CD are much lower than for WD and MF players. 

These findings may help coaches to identify which players would be able to respond to the physical demands during matches played with 3DP tactical formations. Also, from the perspective of football-specific conditioning, it is important to use these findings when creating position-specific training drills for players on specific playing positions.” 

Reviewer 3 Report

Dear authors,

This manuscript investigates running performances of specific positions of elite soccer players in two formations (playing with 3 or 4 defenders). While I think the importance of conducting such a research to quantify and analyze data is crucial; however, translating the data into practice in order to help coaching personal and health partitioners to improve athletic performance and health should be clearly laid out in the manuscript. Therefore, I have several comments on introduction/methods/discussion to help the authors to improve the quality of the manuscript.

Abstract-

Line 17 Please consider removing “either” as you have stated more than 2 groups. This comment applies to the entire manuscript.

Line 24- You have already presented an abbreviation for “running performance” so you might as well stick with throughout the manuscript.

Line 25-27 “The characteristics and differences shown in this study may provide useful information for coaching staff regarding changing in-season tactical formations. Additionally, the results are useful for optimizing training programs for football players with different playing positions.”

If I am to only read the abstract, I won’t get any practical message out of this study. Please consider being more specific in your conclusion in the abstract. The above statements are too general and not well applicable to your research.

Introduction

Line 53-57 “ Although providing important information about possible associations between tactical solutions and running performances in football, the previously cited studies have investigated team tactical formations with four defensive players (4DP), which is undoubtedly the most common tactical formation in modern football; however, we are witnessing the growing trend of using tactical formations with three defensive players (3DP) [19].”

The above sentences is too long and confusing; please consider splitting it into separate sentences.

Materials and Methods

Please state the exact number of players who have participated in this project.

Line 85 “…and all of them signed an informed consent form agreeing to participate in the study. Player…”

 Please consider removing “from agreeing”.

Line 89- “ 3DP tactical formations were used in 10 games and included the 3-5-2 and

3-4-1-2 systems (Figure 1).”

Please stick either with formations or systems. In this case, I would suggest replacing systems to formations.

Line 115 “Apart from the tactical formations (3DP vs. 4 DP) and playing positions (CD, WD, MF, and FW),”

This should be omitted as it is not necessary.

Line 126 Please add a line stating that you have used SPSS software for statistical analysis.

Discussion

Line 203-204 “To defend successfully, CDs must return to their starting position in defence while running at maximal and submaximal intensities.”

It doesn’t make sense here to say submaximal intensities when players are trying to return to their starting position for defense. Please consider removing “submaximal”.

Line 239-241 Among MFs, higher values for almost all RP metrics were determined for 3DP tactical formations; however, the total number of accelerations and decelerations differentiate between the 3DP and 4DP formations to a great extent.

This sentence is incomplete.

Conclusions

Line 309-310 …”coaches can be sure that FW players will be able to respond to the

physical demands of their position, regardless of the number of players in defence or midfield.”

I don’t think we can say that “ coaches can be sure”… According to what has been acknowledged as limitations of this project, I would argue the strength of the above statement. Perhaps try to tone down the language.

Author Response

Reviewer 3

Comments and Suggestions for Authors

Dear authors,

This manuscript investigates running performances of specific positions of elite soccer players in two formations (playing with 3 or 4 defenders). While I think the importance of conducting such a research to quantify and analyze data is crucial; however, translating the data into practice in order to help coaching personal and health partitioners to improve athletic performance and health should be clearly laid out in the manuscript. Therefore, I have several comments on introduction/methods/discussion to help the authors to improve the quality of the manuscript.

Thank you very much for the finding our manuscript very important. We will try to follow all your comments and amended manuscript accordingly.

Abstract-

Line 17 Please consider removing “either” as you have stated more than 2 groups. This comment applies to the entire manuscript.

We accepted this suggestion and amended manuscript accordingly. 

Line 24- You have already presented an abbreviation for “running performance” so you might as well stick with throughout the manuscript.

We agree with you. All abbreviation throughout the manuscript are now corrected.

Line 25-27 “The characteristics and differences shown in this study may provide useful information for coaching staff regarding changing in-season tactical formations. Additionally, the results are useful for optimizing training programs for football players with different playing positions.”

If I am to only read the abstract, I won’t get any practical message out of this study. Please consider being more specific in your conclusion in the abstract. The above statements are too general and not well applicable to your research.

We accepted your suggestion and added practical applications in the abstract sections. Text now reads:

“Abstract: Running performances (RPs) are known to be important parameters of success in football (soccer), but there is a lack of studies where RPs are contextualized regarding applied tactical solutions. This study aims to quantify and analyse the differences in position-specific RPs in professional football, when games are played with three defensive players (3DP) and four defensive players (4DP). The participants here include professional football players (M ± SD, age 23.57 ± 2.84 years, body height 181.9 ± 5.17 cm, body mass 78.36 ± 4.18 kg) playing at the highest competitive level in Croatia. RPs were measured by global positioning system and classified into four groups based on playing positions: central defenders (CD; n = 47), wide defenders (WD; n = 24), midfielders (MF; n = 48), or forwards (FW; n = 19). Analysis of variance and discriminant canonical analysis are used to identify differences between 3DP and 4DP tactical solutions in terms of the RPs for each playing position. The number of accelerations and decelerations most significantly contributed to the differentiation of 3DP and 4DP among MFs (Wilks Lambda = 0.31, p < 0.001), with higher occurrences with 3DP. For CDs, total distance, and high-intensity running were higher in 3DP (Wilks Lambda = 0.66, p < 0.001). No multivariate differences were found for FW and WD players in terms of the RPs between 3DP and 4DP tactical formations. The characteristics and differences shown in this study may provide useful information for coaching staff regarding changing in-season tactical formations. Additionally, the results are useful for optimizing training programs for football players with different playing positions. When changing from 4DP to 3DP tactical formations WDs training programs should include more of high-intensity running, while MFs training programs should be more based on short intensity activities (accelerations and decelerations).

Keywords: team sports; match; tactics; physical capacities; strength and conditioning”

Introduction

Line 53-57 “ Although providing important information about possible associations between tactical solutions and running performances in football, the previously cited studies have investigated team tactical formations with four defensive players (4DP), which is undoubtedly the most common tactical formation in modern football; however, we are witnessing the growing trend of using tactical formations with three defensive players (3DP) [19].”

The above sentences is too long and confusing; please consider splitting it into separate sentences.

We agree with you and split this part into the separate sentences. Text now reads:

“Although providing important information about possible associations between tactical solutions and running performances in football, the previously cited studies have investigated team tactical formations with four defensive players (4DP). 4DP tactical formations are undoubtedly the most common tactical formations in modern football. However, we are witnessing the growing trend of using tactical formations with three defensive players (3DP) [21].”

Materials and Methods

Please state the exact number of players who have participated in this project.

Exact number of players that participated in this project is added (n = 20) in the first paragraph of Methods. Text reads:

“The participants of this study (n = 20) were professional football players (M ± SD, age 23.57 ± 2.84 years, body height 181.9 ± 5.17 cm, body mass 78.36 ± 4.18 kg) and all of them signed an informed consent form agreeing to participate in the study.”

Line 85 “…and all of them signed an informed consent form agreeing to participate in the study. Player…”

Please consider removing “from agreeing”.

Amended accordingly. Thank you.

Line 89- “ 3DP tactical formations were used in 10 games and included the 3-5-2 and

3-4-1-2 systems (Figure 1).”

Please stick either with formations or systems. In this case, I would suggest replacing systems to formations.

Amended accordingly. Thank you.

 Line 115 “Apart from the tactical formations (3DP vs. 4 DP) and playing positions (CD, WD, MF, and FW),”

This should be omitted as it is not necessary.

Thank you for this suggestion. We deleted this part.

Line 126 Please add a line stating that you have used SPSS software for statistical analysis.

Thank you for this suggestion. We used Statistica 13.0 software, so we added next sentence:

“For all analyses, Statistica 13.0 (TIBCO Software Inc., Greenwood Village, CO, USA) was used, and a p < 0.05 was applied.”

Discussion

Line 203-204 “To defend successfully, CDs must return to their starting position in defence while running at maximal and submaximal intensities.”

It doesn’t make sense here to say submaximal intensities when players are trying to return to their starting position for defense. Please consider removing “submaximal”.

Basically, players do not always achieve maximal sprint when coming back into the defence. Sometimes they reach maximal intensity (sprinting distance; over 25.2 km/h) and sometimes submaximal intensities (high speed distance; between 19.8 and 25.2 km/h).

However, we accepted your suggestion and amended this part of the text to cleared out what we thought. Text now reads:

“To defend successfully, CDs must return to their starting position in defence. While running back, maximal and submaximal intensities are typically reached. This includes high-intensity running (i.e., high-speed running and sprinting).”

Line 239-241 Among MFs, higher values for almost all RP metrics were determined for 3DP tactical formations; however, the total number of accelerations and decelerations differentiate between the 3DP and 4DP formations to a great extent.

This sentence is incomplete.

Thank you for noticing this. Sentence is deleted according to you comment and according to comments of other reviewers.

Conclusions

Line 309-310 …”coaches can be sure that FW players will be able to respond to the physical demands of their position, regardless of the number of players in defence or midfield.”

I don’t think we can say that “ coaches can be sure”… According to what has been acknowledged as limitations of this project, I would argue the strength of the above statement. Perhaps try to tone down the language.

Thank you for this suggestion. Conclusion is systematically re-written, so this part of the text is moved to the Practical applications section. However, we followed your suggestion and toned down this sentence. Text now reads:

“In particular, if a change of tactical formations does not include a change in the number of players in attack, FW players will probably be able to respond to the physical demands of their position, regardless of the number of players in defence or midfield.”